# A proof-of-concept study on the genomic evolution of Sars-Cov-2 in molnupiravir-treated, paxlovid-treated and drug-naïve patients

Claudia Alteri[1,2], Valeria Fox[1,2,6], Rossana Scutari[1,2,6], Giulia Jole Burastero[3], Sara Volpi[3], Matteo Faltoni[3], Vanessa Fini[1], Annarita Granaglia[1], Sara Esperti[3], Altea Gallerani[3], Valentino Costabile [1], Beatrice Fontana[3], Erica Franceschini[4], Marianna Meschiari [4], Andrea Campana[5], Stefania Bernardi[5], Alberto Villani [5], Paola Bernaschi[1], Cristina Russo[1], Giovanni Guaraldi [3], Cristina Mussini[3] & Carlo Federico Perno [1✉]

Little is known about SARS-CoV-2 evolution under Molnupiravir and Paxlovid, the only antivirals approved for COVID-19 treatment. By investigating SARS-CoV-2 variability in 8 Molnupiravir-treated, 7 Paxlovid-treated and 5 drug-naïve individuals at 4 time-points (Days 0-2-5-7), a higher genetic distance is found under Molnupiravir pressure compared to Paxlovid and no-drug pressure (nucleotide-substitutions/site mean±Standard error: $18.7 \times 10^{-4} \pm 2.1 \times 10^{-4}$ vs. $3.3 \times 10^{-4} \pm 0.8 \times 10^{-4}$ vs. $3.1 \times 10^{-4} \pm 0.8 \times 10^{-4}$, $P = 0.0003$), peaking between Day 2 and 5. Molnupiravir drives the emergence of more G-A and C-T transitions than other mutations ($P = 0.031$). SARS-CoV-2 selective evolution under Molnupiravir pressure does not differ from that under Paxlovid or no-drug pressure, except for orf8 (dN > dS, $P = 0.001$); few amino acid mutations are enriched at specific sites. No RNA-dependent RNA polymerase (RdRp) or main proteases (Mpro) mutations conferring resistance to Molnupiravir or Paxlovid are found. This proof-of-concept study defines the SARS-CoV-2 within-host evolution during antiviral treatment, confirming higher in vivo variability induced by Molnupiravir compared to Paxlovid and drug-naive, albeit not resulting in apparent mutation selection.

[1] Multimodal Research Area, Microbiology and Diagnostics of Immunology Unit, Bambino Gesù Children Hospital IRCCS, Rome, Italy. [2] Department of Oncology and Hemato-Oncology, University of Milan, Milan, Italy. [3] Department of Infectious Diseases, University of Modena and Reggio Emilia, Modena, Italy. [4] Department of Infectious Diseases, AOU Modena, Modena, Italy. [5] Academic Department of Pediatrics, Bambino Gesù Children's Hospital IRCCS, Rome, Italy. [6] These authors contributed equally: Valeria Fox, Rossana Scutari. ✉email: carlofederico.perno@opbg.net

Two oral small antiviral drugs have so far emergency-use authorized by FDA, EMA and other regulatory authorities around the world for the treatment of Coronavirus Disease 2019 (COVID-19) in adults who do not need supplemental oxygen and who are at high risk of progressing to severe COVID-19: Molnupiravir and Paxlovid (https://www.fda.gov/news-events/press-announcements/coronavirus-covid-19-update-fda-authorizes-additional-oral-antiviral-treatment-covid-19-certain, https://www.fda.gov/news-events/press-announcements/coronavirus-covid-19-update-fda-authorizes-first-oral-antiviral-treatment-covid-19).

Molnupiravir is a bioactive isopropylester prodrug of the ribonucleoside analogue β-D-N4-hydroxycytidine (NHC, EIDD1931) used as a broad-spectrum antiviral agent and originally developed against alphaviruses. Following administration, Molnupiravir is converted into active NHC in the plasma by host esterases and is then intracellularly phosphorylated by host kinases, leading to its triphosphate form (NHC-TP)[1]. In this form, it can act as an alternate and competitive substrate for the SARS-CoV-2 RNA-dependent RNA polymerase (RdRp), interfering with viral replication. Similar to other mutagenic nucleotides, NHC-TP can affect base pairing in various ways, due to the existence of different tautomeric forms: in its hydroxylamine form, NHC-TP acts like cytosine (C), thus leading to base pairing with guanine (G), while in its oxime form acts like uracil (U), leading to adenine (A) base pairing. During the sequential steps of RNA replication, the RdRp incorporates NHC-TP into the negative or positive sense genomic RNA (±gRNA) of SARS-CoV-2 as NHC-monophosphate (NHC-MP)[2,3]. When present as NHC-MP, both tautomeric forms coexist, resulting in the incorporation of GTP and ATP, causing an increase in G to A and C to U transitions in the final +gRNA template. As the viral RNA genome is amplified in the cell, the effect of this accumulation of mutations comprise mutagenic transcription and translation products, ultimately yielding non-functional genomes, in a process which is known as lethal mutagenesis or error catastrophe[4]. Differently from what happens for the nucleoside analogue Remdesivir, whose translocation barrier can in part be eliminated by the proofreading activity of the Nsp14 exonuclease[5], the mutations induced by Molnupiravir are not corrected, due to the stability of the M:G and M:A base pairs formed by the NHC-MP[6], thus resulting in higher antiviral activity.

Paxlovid (PF-07321332) is a co-packaged combination of Nirmatrelvir, a novel main protease (Mpro) inhibitor, specifically designed to block the activity of the SARS-CoV-2 Mpro, and Ritonavir, a strong cytochrome P450 (CYP) 3A4 inhibitor[7,8]. Paxlovid works intracellularly by binding to the highly conserved main protease of the SARS-CoV-2. By this mechanism, Paxlovid inhibits viral replication at the polyprotein maturation step, which occurs before viral RNA replication.

Data resulting from clinical trials have confirmed both Molnupiravir and Paxlovid antiviral efficacy and tolerability, highlighting their efficacy in reducing hospitalization and death in non-hospitalized adults with mild-to-moderate COVID-19, and reporting the absence of any major adverse effects[9–11].

From a virological point of view, a significant decrease in the time required for viral RNA clearance and a higher proportion of overall viral RNA clearance was observed in participants treated with Molnupiravir compared to those not treated[12]. Similar data are predicted for Paxlovid[13].

So far, limited information is available about Molnupiravir and Paxlovid mutagenic potential in patients, as well as about the risk of drug resistance emergence. It has been proposed that Molnupiravir could have a high genetic barrier to resistance[14,15], due to the random accumulation of mutations throughout the genome, together with its short-term use, which suggests a low likelihood for the appearance of resistance as recently confirmed by the AGILE phase IIa clinical trial[16]. Regarding Paxlovid, cell culture systems revealed the selection of Nirmatrelvir resistant Mpro variants characterized by high fitness and by weakness in Nirmatrelvir-Mpro binding[17–19].

In such a rapidly evolving pandemic, real-world data are now demanded to understand the SARS-CoV-2 evolution under small antiviral agents' pressure. In this light, this proof-of-concept study defines the dynamics of the evolution of SARS-CoV-2 genome in infected individuals across the five days of small antiviral drug treatment (Molnupiravir or Paxlovid), focusing the attention on the emergence of single nucleotide polymorphisms (SNPs) in the whole genome, and their persistence after the end of treatment (Day 7). Results are compared with the SARS-CoV-2 evolution observed in drug-naïve individuals.

## Results

**Samples characteristics.** From March 2022 to May 2022, nasopharyngeal swabs at four different time points (Day 0, Day 2, Day 5, and Day 7) were available from 19 adult subjects treated with either Molnupiravir or Paxlovid at the Azienda Ospedaliero-Universitaria Policlinico of Modena, and from 5 drug-naïve pediatric subjects at the Bambino Gesù Children Hospital IRCCS in Rome. Eleven individuals (55.0%) were male, and 18 (90.0%) had at least one comorbidity. For both Molnupiravir and Paxlovid, treatment was started at Day 0 and ended at Day 5. Complete follow-up or SARS-CoV-2 whole genome sequencing at each time point was available for 20 individuals (8 for Molnupiravir, 7 for Paxlovid and 5 for drug-naïve) (Supplementary Fig. 1). Samples characteristics are reported in Table 1.

Whole-genome sequencing analysis revealed that all SARS-CoV-2 infections belonged to Omicron Clade, BA.2 sublineage. Half of them ($N = 10$, 50.0%) belonged to lineage BA.2.9, followed by pure BA.2 ($N = 4$, 20.0%), BA.2.12/BA.2.12.1 ($N = 3$, 15.0%) and BA.2.3/BA.2.3.15 ($N = 3$, 15.0%). Median (Interquartile range, IQR) date of symptoms' start, and first positivity were 2 May 2022 (27 April 2022–12 May 202) and 02 May 2022 (28 Apr 2022–11 May 2022), respectively.

Among the 3 study groups, age and prevalence of comorbidities were higher in Molnupiravir and Paxlovid treated individuals compared to drug-naïve ($P = 0.005$ and 0.032). Molnupiravir-treated and Paxlovid-treated patients had a first positivity date more recent compared to drug-naïve individuals ($P = 0.033$), and the prevalence of the sublineage BA.2.9 was more prevalent in Molnupiravir-treated and drug-naïve individuals compared to Paxlovid-treated patients ($P = 0.002$) (Table 1).

**SARS-CoV-2 RNA decay.** Looking at SARS-CoV-2 load decay across time points in the three study groups, Molnupiravir-treated patients displayed a trend of greater viral load decrease from Day 0 to Day 7 with respect to Paxlovid-treated and drug-naïve individuals (median [IQR] −4.3 [−2.4;−6.2] vs. −3.0 [−1.6;−3.7] vs. −2.5 [−1.1;−3.2] $\log_{10}$ copies/mL respectively, $P = 0.056$), highlighting a more rapid decay kinetics with nucleoside analogue than what seen in Paxlovid-treated and drug-naïve individuals (Fig. 1).

The viral load observed at each time point was summarized below. At the beginning of Molnupiravir and Paxlovid treatment (Day 0), median SARS-CoV-2 nasopharyngeal load was 7.2 (IQR: 6.5–7.7) log copies/mL, and no significant differences were observed among Molnupiravir-treated, Paxlovid-treated and drug-naïve individuals (Table 1 and Fig. 1).

At Day 2 of treatment, median SARS-CoV-2 load decreased to 5.1 (IQR: 4.0–6.3) $\log_{10}$ copies/mL without significant differences among Molnupiravir-treated, Paxlovid-treated and drug-naïve individuals, respectively (Table 1 and Fig. 1).

At Day 5 of treatment, median SARS-CoV-2 nasopharyngeal load was 4.3 (IQR: 3.5–6.0) $\log_{10}$ copies/mL, and significant lower viral

**Table 1 Samples' characteristics of the 20 individuals enrolled in the study against study groups.**

| | Overall N = 20 | Study group | | | p-value |
| --- | --- | --- | --- | --- | --- |
| | | Molnupiravir treated N = 8 | Paxlovid treated N = 7 | Drug Naïve N = 5 | |
| Sex, Male[a] | 11 (55.0) | 5 (62.5) | 3 (42.8) | 3 (60.0) | 0.723 |
| Age (years)[a] | 51 (18–62) | 58 (51–69) | 61 (46–63) | 7 (2–11) | 0.005 |
| Presence of at least one comorbidity[a] | 18 (90.0) | 8 (100.0) | 7 (100.0) | 3 (60.0) | 0.036 |
| Start of Symptoms[a] | 02 May 2022 (27 Apr 2022–12 May 2022) | 02 May 2022 (02 May 2022–12 May 2022) | 30 Apr 2022 (27 Apr 2022–26 May 2022) | 01 Apr 2022 (23 March 2022–21 Apr 2022) | 0.150 |
| First positivity[a] | 02 May 2022 (28 Apr 2022–11 May 2022) | 03 May 2022 (02 May 2022–12 May 2022) | 01 May 2022 (29 Apr 2022–27 May 2022) | 01 Apr 2022 (26 March 2022–28 Apr 2022) | 0.033 |
| Lineages;[a] | | | | | |
| BA.2 | 4 (20.0) | 1 (12.5) | 2 (28.6) | 1 (20.0) | 0.739 |
| BA.2.9 | 10 (50.0) | 7 (87.5) | 0 (0.0) | 3 (60.0) | 0.002 |
| BA.2.12/BA.2.12.1 | 3 (15.0) | 0 (0.0) | 3 (42.8) | 0 (0.0) | 0.047 |
| BA.2.3/BA.2.3.15 | 3 (15.0) | 0 (0.0) | 2 (28.6) | 1 (20.0) | 0.283 |
| SARS-CoV-2 RNA at Day 0[b] | 7.2 (6.5–7.7) | 7.1 (6.5–7.3) | 7.2 (6.5–7.7) | 7.7 (7.4–8.0) | 0.134 |
| SARS-CoV-2 RNA at Day 2[b] | 5.1 (4.0–6.3) | 4.8 (3.4–5.8) | 5.4 (4.2–5.8) | 6.5 (5.0–7.9) | 0.193 |
| SARS-CoV-2 RNA at Day 5[b] | 4.3 (3.5–6.0) | 3.6 (1.6–4.5) | 5.4 (3.5–6.0) | 6.4 (5.1–7.0) | 0.008 |
| SARS-CoV-2 RNA at Day 7[b] | 4.0 (0.0–5.5) | 3.1 (0.0–4.5) | 4.8 (3.5–5.4) | 5.5 (3.4–6.6) | 0.083 |

Data are expressed as median (IQR), or N (%).
[a]Two-sided P-values were calculated by Kruskal–Wallis test, or Chi-square test, as appropriate.
[b]Two-sided P-values were calculated by Jonckheere-Terpstra test. SARS-CoV-2 RNA was expressed as log10 copies/mL.

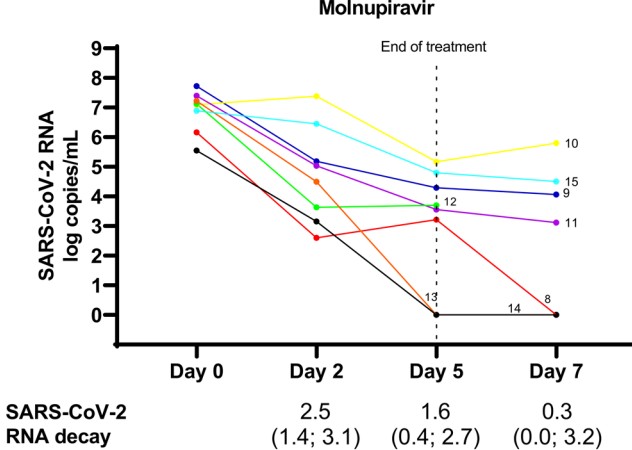

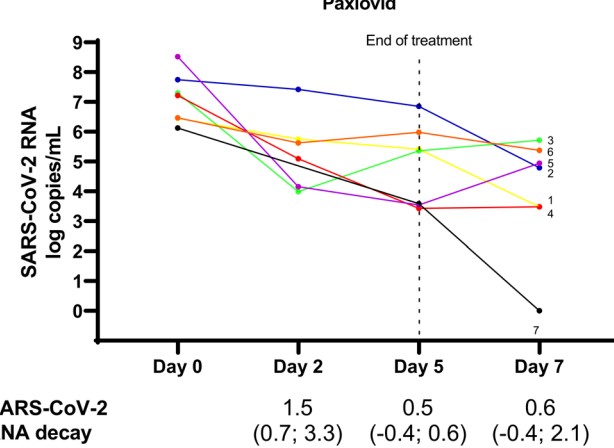

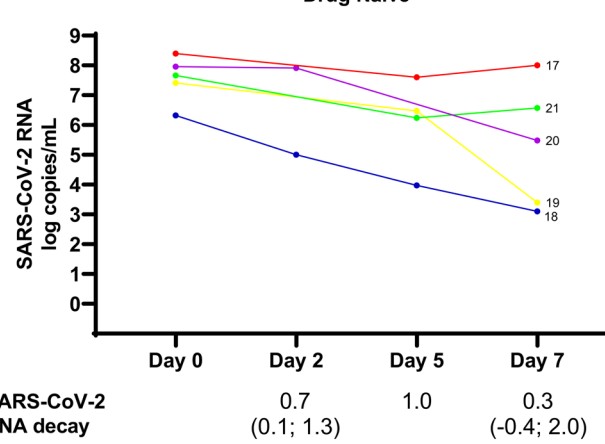

**Fig. 1 SARS-CoV-2 RNA across time points in Molnupiravir-treated, Paxlovid-treated and Drug Naïve patients.** Each patient is depicted by lines of different colors. Day 2 is missing for ID7, ID17, and ID21. Day 5 is missing for ID20. Day 7 is missing for ID12 and ID13. SARS-CoV-2 RNA decay from Day 0 to Day 2, from Day 2 to Day 5 and from Day 5 to Day 7 is expressed as log median (IQR).

load amounts were found in Molnupiravir-treated with respect to Paxlovid-treated and drug-naïve individuals (median [IQR] 3.6 [1.6–4.5] vs. 5.4 [3.5–6.0] vs. 6.4 [5.1–7.0] log$_{10}$ copies/mL, $P = 0.008$) (Table 1 and Fig. 1).

At Day 7 (two days after the end of treatment), median SARS-CoV-2 nasopharyngeal load was 4.0 (IQR: 0.0–5.5 log$_{10}$ copies/mL), and a trend of lower SARS-CoV-2 amounts was detected in Molnupiravir-treated compared to Paxlovid-treated and drug-naïve individuals (median [IQR] 3.1 [0.0–4.5] vs. 4.8 [3.5–5.4] vs. 5.5 [3.6–6.6] log$_{10}$ copies/mL, $P = 0.083$) (Table 1 and Fig. 1).

**Within-host SARS-CoV-2 genetic distance and correlation with treatment.** SARS-CoV-2 sequences obtained at different time points from Molnupiravir-treated, Paxlovid-treated and drug-naïve individuals were subjected to phylogenetic analysis in order to define the SARS-CoV-2 evolution across time points and against treatment (Fig. 2).

ML tree first revealed that all sequences belonging to the same patient correctly clustered together, without intermixing, excluding cross-contamination among samples (Fig. 2). ML tree also confirmed that all sequences belonged to Omicron clade, sublineage BA.2.

Overall, the mean (±SE) within-host SARS-CoV-2 genetic distance was $9.2 \times 10^{-4}$ ($\pm 1.3 \times 10^{-4}$) nucleotide substitutions per site. More pronounced SARS-CoV-2 within-host evolution was observed in Molnupiravir-treated with respect to Paxlovid-treated and drug-naïve individuals, as confirmed by the mean within-host genetic distance, which was significantly higher in individuals treated with Molnupiravir ($18.7 \times 10^{-4} \pm 2.1 \times 10^{-4}$) compared to Paxlovid-treated and drug-naive individuals ($3.3 \times 10^{-4} \pm 0.8 \times 10^{-4}$ and $3.1 \times 10^{-4} \pm 0.8 \times 10^{-4}$, respectively) ($P = 0.0003$) (Table 2 and Fig. 2).

The genetic variability induced by Molnupiravir had a peak between Day 2 and Day 5 (median [IQR] genetic distance: $16.2 \times 10^{-4}$ [$8.5–32.9 \times 10^{-4}$], $P = 0.043$), and remained stable between Day 5 and Day 7 (median [IQR]: $20.9 \times 10^{-4}$[$8.5–29.2 \times 10^{-4}$], $P = 0.144$). No significant differences in genetic distances between different time points were observed in the Paxlovid and drug-naïve study groups (Supplementary Table 1 and Fig. 2).

**Number of SNPs against type and treatment.** The higher SARS-CoV-2 genetic variability induced by Molnupiravir with respect to the other two study groups, was well represented by the higher number of SNPs observed across time points in Molnupiravir-treated (Fig. 3a) compared to Paxlovid-treated (Fig. 3b) and drug-naïve individuals (Fig. 3c).

According to its mechanism of action, Molnupiravir drove the appearance of more G to A and C to T transitions (Fig. 3a, SNPs in red and orange, respectively) than other nucleotide substitutions (Fig. 3a, SNPs in green) (median number [IQR]: 18 [9–40] vs 7 [5–15], $P = 0.031$), irrespectively of time points and SARS-CoV-2 proteins (Supplementary Fig. 2).

Regardless of SNPs type, and in line with what was observed for the genetic distance, the SNPs induced by Molnupiravir started to appear at Day 2 (median [IQR]: 24 [4–34]), increased at Day 5 (45 [21–68]) and remained stable at Day 7 (45 [24–64]) ($P = 0.110$) (Fig. 3a).

During Paxlovid treatment and in the absence of drug pressure, no differences were found in the number of SNPs across time points (median [IQR] at Day 2, 5, 7: 7 [3–9] vs. 9 [1–12] vs. 8 [4–10], $P = 0.600$ for Paxlovid and 8 [2–14] vs. 5 [0–9] vs. 2 [0–4], $P = 0.297$ for drug-naïve) (Fig. 3b, c). No differences in the number of G to A and C to T transitions compared to other mutations were also found ($P > 0.05$). Thus, the variability of the viral genome in Paxlovid-treated patients is similar to the natural variability observed in not-treated patients, and far lower than that found during Molnupiravir treatment.

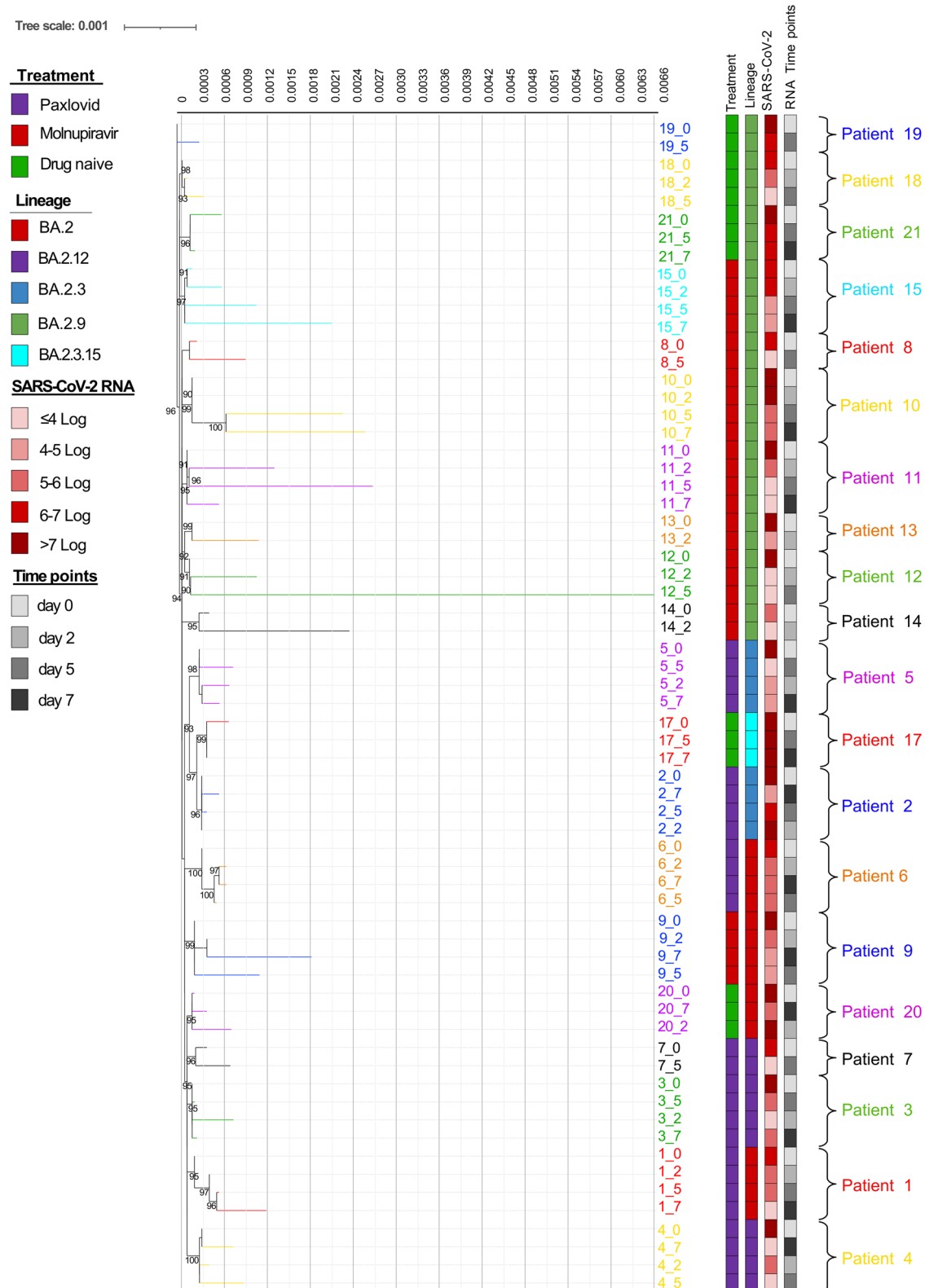

**Fig. 2 Estimated maximum likelihood phylogenetic tree of the SARS-CoV-2 strains from Molnupiravir-treated, Paxlovid-treated and drug naïve individuals.** The phylogeny was estimated with IqTree2 using the best-fit model of nucleotide substitution TN + F + G4 with 1000 replicates fast bootstrapping. Leaves number represents the sample ID, bootstrap values higher than 90 are shown on branches, color of branches represents the different patients whose ID is also reported next to the tree with brackets. Information regarding treatment, lineage, SARS-CoV-2 viral load and the time points analyzed were reported in the annotation columns in the right-hand side of the figure. Tree scale is expressed as nucleotide substitutions per site.

**Table 2 Within-host genetic distance according to study groups.**

| Study group | Patient | Within-host genetic distance | | Within-host genetic distance according to study's group | | $P$-value[a] |
|---|---|---|---|---|---|---|
| | | Mean | Standard error | Mean | Standard error | |
| Molnupiravir | 8 | 0.00180 | 0.00021 | $18.7 \times 10^{-4}$ | $2.1 \times 10^{-4}$ | 0.0003 |
| | 9 | 0.00085 | 0.00019 | | | |
| | 10 | 0.00078 | 0.00021 | | | |
| | 11 | 0.00209 | 0.00028 | | | |
| | 12 | 0.00151 | 0.00017 | | | |
| | 13 | 0.00187 | 0.00020 | | | |
| | 14 | 0.00486 | 0.00025 | | | |
| | 15 | 0.00081 | 0.00014 | | | |
| Paxlovid | 1 | 0.00045 | 0.00011 | $3.3 \times 10^{-4}$ | $0.8 \times 10^{-4}$ | |
| | 2 | 0.00029 | 0.00006 | | | |
| | 3 | 0.00039 | 0.00009 | | | |
| | 4 | 0.00002 | 0.00002 | | | |
| | 5 | 0.00027 | 0.00007 | | | |
| | 6 | 0.00061 | 0.00011 | | | |
| | 7 | 0.00022 | 0.00008 | | | |
| Drug naive | 17 | 0.00034 | 0.00012 | $3.1 \times 10^{-4}$ | $0.8 \times 10^{-4}$ | |
| | 18 | 0.00048 | 0.00009 | | | |
| | 19 | 0.00020 | 0.00007 | | | |
| | 20 | 0.00028 | 0.00007 | | | |
| | 21 | 0.00025 | 0.00007 | | | |

Genetic distance is expressed as nucleotide substitutions per site.
[a]Two-sided $P$-value was calculated by Jonckheere–Terpstra test.

**Positive selection and missense amino acid mutations**. All the 33 SARS-CoV-2 proteins evolved under a neutral selection ($P > 0.05$) across study groups, except for three proteins that seemed to evolve under a purifying selection in Molnupiravir-treated individuals ($P \leq 0.05$) (Table 3). A negative selection (dN<dS) was observed in the structural protein N ($P = 0.05$) and the non-structural protein nsp9 ($P = 0.03$), known to be involved in replication-transcription complex and autophagosome inhibition[20–23] (Table 3). The negative selection characterizing the evolution of these proteins during Molnupiravir treatment probably reflects the low number of non-synonymous mutations detected across time points. In particular, 17/33 (51.5%) and 4/11 (36.4%) SNPs observed in N and nsp9 induced missense mutations. None of these SNPs was detected in more than one time point, and none of the mutated codons was under purifying selection.

Orf8, known to be involved in immune response evasion[24], seemed to be the only protein evolving under a positive selection during Molnupiravir treatment (dN>dS, $P = 0.001$), due to the presence of 7/7 (100%) non-synonymous mutations. Among the mutated amino acid positions, none was under purifying selection, and only orf8 A27927G-T12A persisted between Day 5 (intra-patient prevalence: 10.3%) and Day 7 (intra-patient prevalence: 55.8%) of ID10.

When considering RdRp and Mpro, target sites of Molnupiravir and Paxlovid, respectively, no evidence of evolution under positive selection was reported (Table 3). Among the 39 SNPs inducing amino acid changes in the RdRp across time points in Molnupiravir-treated patients, 12 were localized in the fingers, 9 in the interface domain, 5 in the palm, and 3 in the thumb domain (Fig. 4). The remaining mutations were localized in the NiRAN domain and in the hairpin like structure. Among these missense mutations, no evidence of persistence was reported, except for G13604A-R55H appearing at Day 2 (intra-patient prevalence: 5.3%) and persisting at Day 5 (intra-patient prevalence: 14.3%) of ID11. No amino acid change was observed in the RdRp active site residues[25].

Of note, no evidence of extensive missense mutational rate was reported in Mpro during Paxlovid treatment as well as no evidence of so far known (https://www.fda.gov/media/155050/download)[17] Mpro mutations conferring resistance to Paxlovid was found in Paxlovid treated patients. Only the following Mpro mutations were found: C10301T-Q83* found at Day 7 of patient 4 with a relative abundance of 44.6%, and T10504C-F150F found at Day 2 of patient 3 with a relative abundance of 15.1%.

Interestingly, the mutation C10376T-P108S, whose emergence was observed during Nirmatrelvir treatment in SARS-CoV-2 cell culture, was observed at Day 5 of Molnupiravir treatment in patient 11 (relative abundance of 14.5%). However, no data of the reduction of Nirmatrelvir activity was reported for this mutation in biochemical assays (https://www.fda.gov/media/155050/download). Since the loss of the nsp14 could increase the sensitivity of SARS-CoV-2 to Molnupiravir, we further investigate the effect of Molnupiravir in inducing variability on the proofreading function of nsp14. No evidence of evolution under positive selection was reported (Table 3). Among the 23 missense mutations, only two increased their prevalence from 11.1% and 5.4% at Day 5 to 52.7% and 45.3% at Day 7 of ID10 (nsp14 C18452T-A138V and G18520A-V161I). No amino acid change was observed in the nsp14 active site residues[26].

## Discussion

This proof-of-concept study on the SARS-CoV-2 genomic evolution under small molecule antivirals pressure indicates that Molnupiravir induced a higher within-host SARS-CoV-2 variability compared to Paxlovid, finding supported by the higher number of SNPs observed across time points. The SNPs induced by Molnupiravir appeared randomly along the SARS-CoV-2 genome, were most G to A and C to T transitions and showed their peak at Day 5 of treatment, when the genetic distance showed its highest value, and the viral load reached its lowest peak. Two days after the end of treatment (Day 7) the estimates of genetic distance, number of SNPs and viral load were stable compared to those reported at the end of treatment,

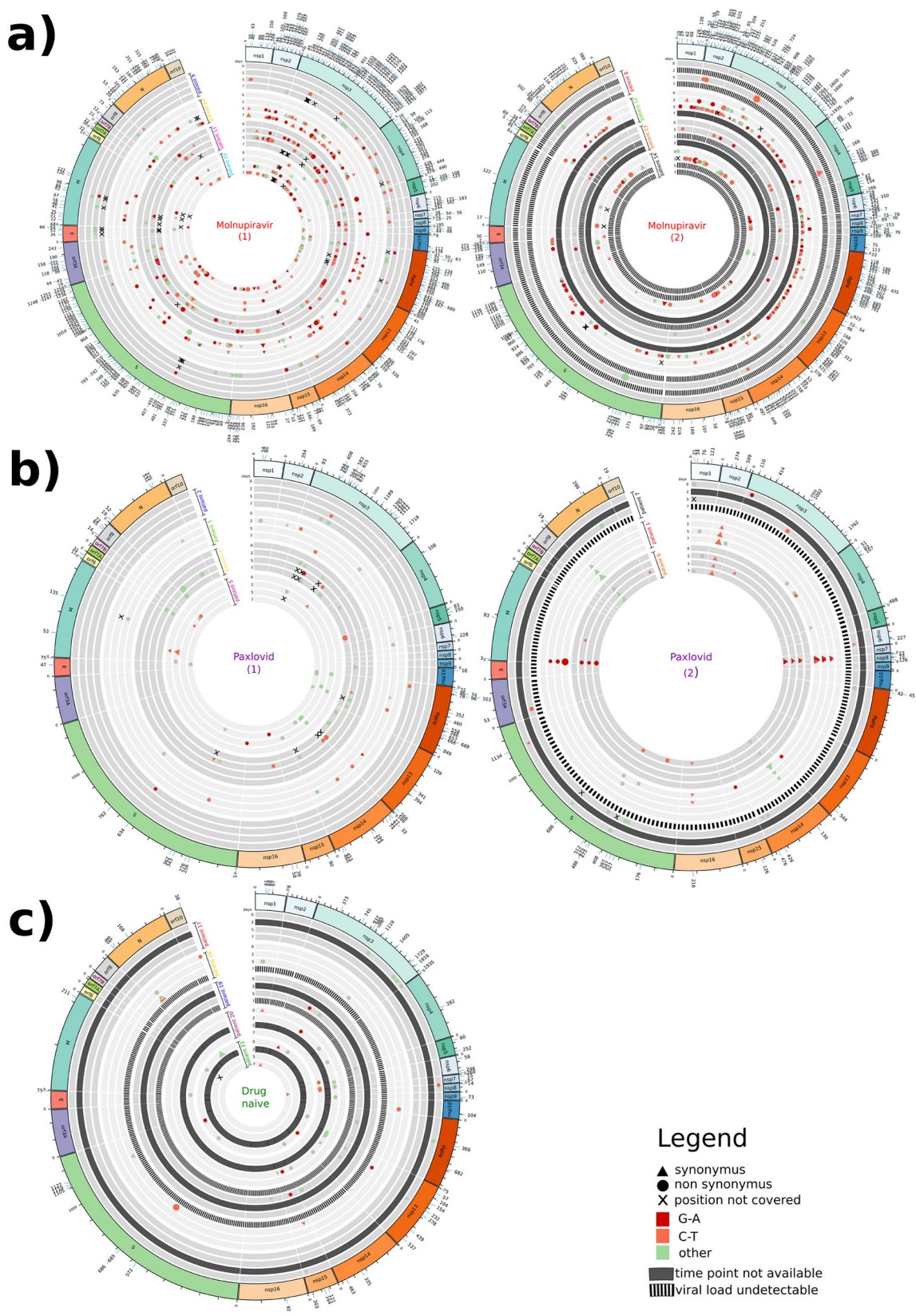

probably reflecting the fact that, even if the treatment was stopped at Day 5, the antiviral drug was still present as a pharmacological tail and continued to drive the genetic evolution.

These findings are consistent with the known error catastrophe induced by Molnupiravir against the virus[4] and are in line with recent large-scale data. In the AGILE phase IIa clinical trial[16], an increase in G to A, C to T and T to C mutations was observed during Molnupiravir treatment, and no amino acid substitutions were enriched consistently at specific sites during the first 5 days of treatment.

Despite the intra-host SARS-CoV-2 variation, already observed among clinical samples throughout the pandemic[27,28], SARS-

**Fig. 3 Circos plots representing the SNPs found in SARS-CoV-2 strains.** Panels represent SARS-CoV-2 strains found in (**a**) Molnupiravir-treated, (**b**) Paxlovid-treated and (**c**) drug-naïve individuals across time points. Each time point is defined by one track and grouped according to patient. Tracks in black represent timepoints not available; striped tracks represent samples not sequenced as viral load was undetectable. SNPs are defined by different colors, shapes and size according to type of nucleotide substitution (red: G-A; orange: C-T; green: other), non-synonymus (circle) or synonymus mutations (triangle), and intra-patient prevalence, respectively. The outside track corresponds to SARS-CoV-2 genome architecture. Overlapping orfs are not reported and comprise: orf2b (coordinates 21744–21860) overlapping S; orf3b (coordinates 25814–25879), orf3c (coordinates 25457–25579), orf3d (coordinates 25524–25694) and orf3d-2 (coordinates 25596–25694) overlapping orf3a; orf9b (coordinates 28284–28574) and orf9c (coordinates 28734–28952) overlapping N.

**Table 3 Codon-based Neutrality for each SARS-CoV-2 gene against study groups.**

| SARS-CoV-2 proteins | Molnupiravir | Paxlovid | Drug naive |
|---|---|---|---|
| nsp1 | $v = -1.4\ p = 0.17$ | $v = -0.2\ p = 0.83$ | $v = -0.7\ p = 0.51$ |
| nsp2 | $v = -1.4\ p = 0.18$ | $v = -1.0\ p = 0.30$ | $v = -1.4\ p = 0.17$ |
| nsp3 | $v = -1.4\ p = 0.17$ | $v = 0.2\ p = 0.85$ | $v = 1.8\ p = 0.08$ |
| nsp4 | $v = -1.2\ p = 0.22$ | $v = -0.1\ p = 0.94$ | $v = 0.9\ p = 0.14$ |
| Mpro (nsp5) | $v = -0.6\ p = 0.56$ | $v = -1.4\ p = 0.18$ | $v = -0.7\ p = 0.51$ |
| nsp6 | $v = -1.3\ p = 0.21$ | $v = -1.5\ p = 0.13$ | $v = 1.9\ p = 0.07$ |
| nsp7 | $v = 1.1\ p = 0.28$ | $v = 0.0\ p = 1.00$ | $v = -0.7\ p = 0.50$ |
| nsp8 | $v = -0.7\ p = 0.50$ | $v = -1.6\ p = 0.12$ | $v = 1.1\ p = 0.29$ |
| **nsp9** | $\mathbf{v = -2.2\ p = 0.03}$ | $v = 0.0\ p = 1.00$ | $v = 1.1\ p = 0.29$ |
| nsp10 | $v = -1.4\ p = 0.18$ | $v = -1.0\ p = 0.30$ | $v = 1.0\ p = 0.30$ |
| nsp11 | $v = 0.0\ p = 1.00$ | $v = 0.0\ p = 1.00$ | $v = 0.0\ p = 1.00$ |
| RdRp (nsp12) | $v = -1.2\ p = 0.24$ | $v = -1.7\ p = 0.09$ | $v = 1.1\ p = 0.29$ |
| nsp13 | $v = -1.6\ p = 0.12$ | $v = 0.9\ p = 0.33$ | $v = -0.7\ p = 0.47$ |
| nsp14 | $v = -1.2\ p = 0.23$ | $v = -1.7\ p = 0.10$ | $v = -1.2\ p = 0.24$ |
| nsp15 | $v = -2.0\ p = 0.05$ | $v = 1.5\ p = 0.15$ | $v = -1.1\ p = 0.27$ |
| nsp16 | $v = -0.9\ p = 0.39$ | $v = -0.8\ p = 0.41$ | $v = -1.1\ p = 0.29$ |
| S | $V = -1.0\ p = 0.33$ | $v = -0.5\ p = 0.66$ | $v = 0.1\ p = 0.96$ |
| orf2b | $v = 0.5\ p = 0.62$ | $v = 1.1\ p = 0.30$ | $v = 0.0\ p = 1.00$ |
| orf3a | $v = 0.2\ p = 0.81$ | $v = -1.1\ p = 0.30$ | $v = -0.2\ p = 0.84$ |
| orf3b | $v = 0.0\ p = 1.00$ | $v = 1.0\ p = 0.31$ | $v = 1.0\ p = 0.31$ |
| orf3c | $v = -0.8\ p = 0.41$ | $v = -1.0\ p = 0.34$ | $v = -1.0\ p = 0.34$ |
| orf3d | $v = -1.0\ p = 0.34$ | $v = -1.0\ p = 0.33$ | $v = -0.9\ p = 0.36$ |
| orf3d-2 | $v = -0.9\ p = 0.36$ | $v = -1.0\ p = 0.33$ | $v = -0.9\ p = 0.36$ |
| E | $v = -0.6\ p = 0.56$ | $v = 1.5\ p = 0.14$ | $v = 1.0\ p = 0.33$ |
| M | $v = -0.6\ p = 0.58$ | $v = 0.4\ p = 0.71$ | $v = 1.1\ p = 0.29$ |
| orf6 | $v = 0.7\ p = 0.49$ | $v = -0.8\ p = 0.41$ | $v = 0.0\ p = 1.00$ |
| orf7a | $v = 1.1\ p = 0.29$ | $v = 1.0\ p = 0.30$ | $v = -1.1\ p = 0.27$ |
| orf7b | $v = 1.4\ p = 0.16$ | $v = -1.0\ p = 0.33$ | $v = 0.0\ p = 1.00$ |
| **orf8** | $\mathbf{v = 3.2\ p = 0.001}$ | $v = -0.7\ p = 0.51$ | $v = 0.0\ p = 1.00$ |
| orf9b | $v = -1.4\ p = 0.18$ | $v = -0.5\ p = 0.65$ | $v = -0.4\ p = 0.69$ |
| orf9c | $v = -1.6\ p = 0.12$ | $v = 0.0\ p = 1.00$ | $v = -1.0\ p = 0.32$ |
| ***N*** | $\mathbf{v = -2.0\ p = 0.05}$ | $v = -1.0\ p = 0.32$ | $v = 1.0\ p = 0.34$ |
| orf10 | $v = 1.8\ p = 0.07$ | $v = -0.9\ p = 0.35$ | $v = 1.0\ p = 0.32$ |

Codon-based Test of Neutrality for each SARS-CoV-2 protein against study group. The test statistic (dN–dS) is reported as V, where dS and dN are the numbers of synonymous and non-synonymous substitutions per site, respectively. The variance of the difference was computed using the bootstrap method (500 replicates). The probability of rejecting the null hypothesis of strict-neutrality (dN = dS) is shown as P-values (P). P-values less than 0.05 are considered significant at the 5% level and are highlighted. Analyses were conducted using the Nei-Gojobori method. All ambiguous positions were removed for each sequence pair (pairwise deletion option). Evolutionary analyses were conducted in MEGA11.

CoV-2 strains under the Molnupiravir and Paxlovid pressure, as well as in drug-naïve samples, did not seem to experience strong selective evolution. By calculating the probability to evolve under purifying selection of each gene, only orf8, known to be involved in immune response evasion[24], seemed to evolve under a positive selection during Molnupiravir treatment, as a consequence of the 7/7 non-synonymous mutations detected randomly along the protein. Among these, only the orf8 A27927G-T12A persisted between Day 5 and Day 7. No clear information is available about the role of this amino acid mutation in protein stability or functionality, but in silico prediction suggests that this mutation might induce a protein stability change (free energy change, ΔΔG, of −1.5 Kcal/mol compared to wild type)[29]. Further in vitro studies are needed to confirm these in silico findings as well as in general to firmly establish whether and how these substitutions evolve under selection pressure.

We noted that this amino acid change was detected with other 4 amino acid mutations (the nsp3 G5572A-M951I and T8004C-V1762A, nsp14 C18452T-A138V and G18520A-V161I) at Day 5 of the Molnupiravir-treated patient ID10, a renal transplant recipient on tacrolimus-based immunosuppression. This mutational pathway was selected at Day 5 and spread at Day 7 when an increase of genetic distance and a post-treatment increase of viral load were detected. Among the mutations observed, the nsp14 mutations fall in the exonuclease domain, and thus might impair the functionality of the error-correction mechanism, leading to accumulation of mutations and increase of variability. These hypotheses require further validation by in vitro studies and docking simulations. The nsp3 G5572A-M951I mutation falls in the PLpro domain, which plays an essential role in the viral replication and the innate immune evasion[30,31]. If the substitution of methionine to

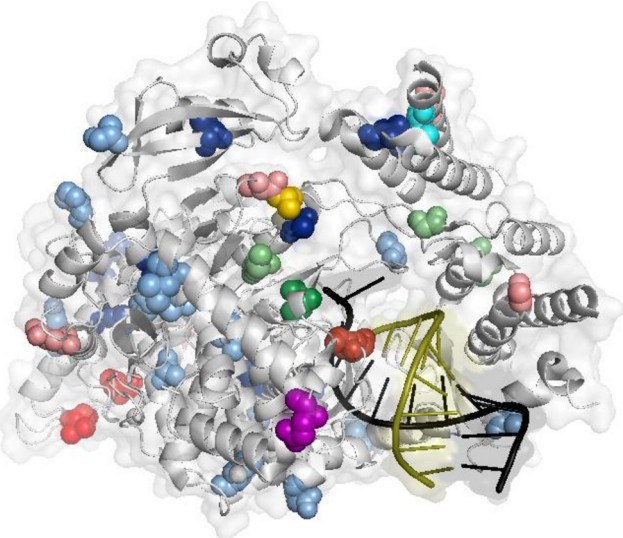

**Fig. 4 RdRp amino acid mutations that emerged during Molnupiravir treatment across time points.** Representation of the structure of the SARS-CoV-2 RdRp with Molnupiravir/NHC complex (PDB:7OZV). Amino acid positions corresponding to the Molnupiravir-induced mutations are shown as spheres of different colors based on day of appearance and relative abundance: (i) mutations appearing at day 2 with relative abundance between 2 and 10% in light green, and between 10 and 20% in green; (ii) mutations appearing at day 5 with relative abundance between 2 and 10% in light blue, and between 10 and 20% in blue; (iii) mutations appearing at day 7 with relative abundance between 2 and 10% in light red, between 10 and 20% in red, and between 20 and 40% in dark red; (iv) mutations appearing at both day 2 and day 5 in cyan (including the G13604A-R55H mutation); (v) mutations appearing at day 2 or day 7 in yellow; vi) mutations appearing at day 5 or day 7 in purple. Template RNA is colored in black, while product RNA in olive. RdRp: RNA-dependent RNA polymerase.

isoleucine in PLpro secondary structure evoked effect on the protein stability, needs to be investigated.

A progressive SARS-CoV-2 genetic variability over time points was observed also in ID12, but with different features. At Day 2 of Molnupiravir-treatment, two nsp14 non-synonymous mutations appeared (C18511A-P158T and C18647T-P203L) with an intra-patient prevalence of 36.0% and 23.7%, respectively. These mutations are both localized in the ExoN domain of nsp14, and the P203L was recently suggested to induce a weakly interaction between nsp14 and nsp10, possibly resulting in reduction of the proofreading activity of nsp14[32]. As a result, nsp14-P203L variants showed higher nucleotide substitution rate at population level, and in our patient induced a pronounced genomic evolution, confirmed by genetic distance $(65.0\ [\pm3.2] \times 10^{-4})$ and emergence of 175 minority SNPs (relative abundance %, median IQR 8.7 [6.5–13.6]) at Day 5.

Understanding the factors underlining the emergence of these mutational pathways in these two patients and whether they are correlated to health status rather than drug pressure might provide essential insights into the risk of SARS-CoV-2 evolutionary mechanisms.

By our proof-of-concept study, no clear evidence of RdRp and Mpro mutations that might contribute to Molnupiravir or Paxlovid resistance (https://www.fda.gov/media/155050/download) was found, even if further insights are needed to define if some of the enriched mutations might be involved in drug resistance mechanisms.

Looking at the variability of the whole spike protein, no site on positive selection or persistent mutation were detected. Six non-synonymous SNPs (V327I, C391F, V401I, T430I, L455S, and R457K) with a relative abundance of median (IQR) 9.8 (6.4–14.9) appeared at different time points in the RBD, but no chance of further selection or spreading was revealed, suggesting no evident risk of SARS-CoV-2 spike evolution under Molnupiravir.

Worthy of mention is the potential impact of the Molnupiravir-induced variability on molecular and Ag diagnostic assays. As already known, SARS-CoV-2 variability in diagnostics targets could hamper the viral load detection or Ag positivity[33]. Regarding the antigenic tests, false negative results might be considered unlikely events under Molnupiravir treatment, because of the rare persistence of non-synonymous mutations and the absence of positive selection in the target genes of antigen tests. With the molecular tests, instead, the risk of detection failure could be due to SNPs falling in the target region of the probes or primers, thus interfering with their binding and thus rendering the assays ineffective, but the multi-target RT-PCR tests used in clinical practice withdraws this risk. These hypotheses require further analyses to be in depth assessed and defined.

Another aspect that should be taken into consideration is the use of Molnupiravir in combination with Paxlovid- or mono-clonal antibodies-based therapy. Since the mode of action of these molecules is different, combined therapy might represent an advantage compared to monotherapy, allowing to boost the efficacy against SARS-CoV-2 infection and to reduce the risk of drug resistance or immune escape mutations. Studies reporting a synergistic effect of Molnupiravir and Paxlovid combination strategy both in vitro[34,35] and in animal models[36] are already published, but ad hoc designed clinical studies will be needed to confirm these findings in humans.

Our study has some limitations. The analysis of negative and positive selection induced by Molnupiravir and Paxlovid should be interpreted carefully, as the number of SNPs that define the purifying selection is small and potential errors introduced during reverse transcription, PCR amplification, or sequencing can slightly impact the results[37]. To overcome these problems, only SNPs having a minimum supporting read frequency of 2% with a depth ≥20, and thus expected to provide more robust results, were retained.

The sample size was relatively small, limiting general considerations. We cannot exclude that denser sampling would reveal novel patterns of mutations not observed here and further address genetic variability. The small sample size might have precluded the possibility to detect drug resistance, whose emergence under small drugs therapies remains a challenge to investigate. The results obtained were mainly addressed to hospitalized and immunocompromised individuals. This population bias may have also caused a substantial underestimation of the viral load decay kinetics and a pronounced genomic variability across time.

In conclusion, this study contributes to define the dynamics of within-host variability of SARS-CoV-2 during Molnupiravir and Paxlovid treatment, confirming the higher genomic variation induced by this nucleoside analogue (as its main mechanism of action also in vivo), and describing in detail the effect of the drug upon the evolution of each different gene constituting the SARS-CoV-2 genome. Molnupiravir induces a number of SNPs that increase against days of treatment. Due to bottleneck or purifying selection, these SNPs appear randomly and rarely persist across time points, limiting the potential risk of selection of viral variants characterized by a fitness advantage during Molnupiravir-treatment. Ad hoc designed studies are necessary to confirm our findings, provide an extensive overview of SARS-CoV-2 intra-host variability and minority variants description during small molecule antivirals treatment, if and how these minority variants can spread in the population, and their potential role in virulence and transmissibility.

## Methods

**Study population**. This retrospective observational proof-of-concept study initially included 24 individuals diagnosed for SARS-CoV-2 infection at Azienda Ospedaliero-Universitaria Policlinico of Modena (Italy) and Bambino Gesù Children Hospital IRCCS of Rome (Italy) from March 2022 to May 2022. Of these, 10 individuals started SARS-CoV-2 antiviral treatment with Molnupiravir, 9 with Paxlovid, and 5 individuals received no treatment (drug-naïve), depending on physicians' discretion.

For each individual with nasopharyngeal swabs available, SARS-CoV-2 viral load and genomic variability were retrospectively analyzed at 4 time points: before SARS-CoV-2 antiviral treatment start (Day 0), 2 days after SARS-CoV-2 antiviral treatment start (Day 2), 5 days after SARS-CoV-2 antiviral treatment start, i.e. end of treatment (Day 5), and 2 days after treatment end (Day 7). The same data were obtained at 4 comparable time points for the 5 drug-naive individuals (Supplementary Fig. 1).

Based on the availability of samples at each time point, results on SARS-CoV-2 viral load quantification and whole-genome SARS-CoV-2 sequencing, 20 individuals were finally selected for the study, and divided in three study groups according to treatment as follows: 8 Molnupiravir-treated, 7 Paxlovid-treated and 5 drug-naïve individuals.

Demographic, epidemiological, and clinical data were obtained retrospectively from pseudonymized electronic medical records.

**Ethics**. The study protocol was approved by local Research Ethics Committee of the Azienda Ospedaliero-Universitaria Policlinico of Modena and Bambino Gesù Children hospitals (prot. 396/2020/OSS/AOUMO and prot. 2384_OPBG_2021). This study was conducted in accordance with the principles of the 1964 Declaration of Helsinki. Informed consent was waived in accordance with the hospital regulations on retrospective observational studies.

**SARS-CoV-2 viral load quantification and sequencing**. Total RNA was extracted from 280 μl of nasopharyngeal swabs (COPAN FLOQSwab with 3 mL of Universal Transport Medium) using QIAamp viral RNA mini kit (Qiagen) following the manufacturer's instruction. SARS-CoV-2 RNA was quantified by means of the QX200™ Droplet Digital™ PCR System (ddPCR, Bio-Rad Laboratories, Inc.) using home-made protocol targeting RdRp and the RNAseP housekeeping gene[38]. Quantitative results were then normalized in number copies/mL of swab.

Whole-genome sequences of SARS-CoV-2 were generated with 50 ng viral RNA template, by using the QIAseq DIRECT SARS-CoV-2 Kit according to the manufacturer's protocol[39,40]. Libraries were then sequenced on a MiSeq instrument (Illumina, San Diego, CA, USA) with 2 × 150-bp paired-end reads. Raw reads were trimmed for adapters and filtered for quality (average q28 threshold and read length >135 nt) using Fastp (v0.20.1)[41]. First and last 15 nucleotides were also removed from all reads. Reference-based assembly was performed with BWA-mem algorithm[42], aligning against the GenBank reference genome NC_045512.2 (Wuhan, collection date: December 2019). SNPs variants were called through a pipeline based on samtools/bcftools[43], and all SNPs having a minimum supporting read frequency of 2% with a depth ≥20 were retained. Consensus was generated using the github freely distributed software vcf_consensus_builder (https://github.com/peterk87/vcf_consensus_builder).

**Genetic distance and Phylogenetic analysis**. SARS-CoV-2 lineages of the SARS-CoV-2 consensus sequences obtained were assigned according to Pangolin application (Pangolin v4.1.1, https://github.com/cov-lineages/pangolin/releases). Sequences were aligned using MAFFT v7.475 and manually inspected using Bioedit. The final alignment comprised 65 sequences of 29,184 nucleotides of length. The within-host viral genetic distance (always expressed as nucleotide substitutions per site) over all time points, and between time points intervals (Day 0-Day 2; Day 2-Day 5; Day 5-Day 7) was calculated with MEGA (v11)[44] under the Tamura-Nei model[45]. Differences in average SARS-CoV-2 genetic distance among the study's groups was then evaluated by Kruskal–Wallis test. Differences in viral genetic distance between time points intervals within each study group were assessed by Wilcoxon signed-rank test.

In order to further explore the within-host and within-group evolution of SARS-CoV-2, a maximum likelihood (ML) phylogeny tree was performed with IqTree2 (v2.1.3)[46] with 1000 bootstrap replicates, using the best-fit model of nucleotide substitution TN + F + G4 inferred by ModelFinder[47]. Annotation of the phylogenetic tree, including information about study group, lineages, SARS-CoV-2 viral load and timepoints, was performed with iTOL[48].

**Non-synonymous and synonymous substitution rate estimation**. We further analyzed genetic variability across all the 33 SARS-CoV-2 genes for evidence of purifying selection or neutral selection. Genes and their overlapping open reading frames (ORFs) were considered according to the reference genome (GenBank accession number NC_045512) and Jungreis et al. 2021[49]. The rates of synonymous (dS) and non-synonymous (dN) mutations per SARS-CoV-2 gene were computed on MEGA (v11), using the Nei-Gojobori method (Proportion)[50], by removing all the constitutive SNPs and using the bootstrap method with 500 replicates. If genes evolve under neutral selection, the SNPs are likely to distribute equally at each

codon position and the difference between non-synonymous substitution and synonymous substitution tends to be not significant ($P > 0.05$). On the contrary, positive (dN>dS) or negative (dN<dS) selection suggests that the SNPs are not equally distributed at each codon position and the difference between non-synonymous substitution and synonymous substitution tends to be significant ($P \le 0.05$). In order to identify single sites under purifying selection ($P \le 0.1$) the Fixed Effects Likelihood (FEL) was applied[51].

**Statistics and reproducibility**. Descriptive statistics are expressed as median values and interquartile range (IQR) for continuous data and number (percentage) for categorical data. To assess significant differences, chi-squared test for trend and Kruskal–Wallis were used for categorical and continuous variables, respectively. Jonckheere-Terpstra test was used to define differences in viral load, viral load decay and SARS-CoV-2 genetic variability at different time points among study groups.

The presence of SNPs at the different time points, together with the information about the type of mutation (synonymous vs non-synonymous), the type of nucleotide substitution (G-A, C-T, or other), and their relative abundance, were graphically represented as circus plots using the Circos package v0.69-9[52]. Statistical analyses were performed with SPSS software package for Windows (version 23.0, SPSS Inc., Chicago, IL). A two-sided p-value <0.05 was considered statistically significant.

**Reporting summary**. Further information on research design is available in the Nature Portfolio Reporting Summary linked to this article.

## Data availability

Nature Research reporting summary was uploaded as related manuscript file. The source data behind Figs. 1, 3, 4 and Supplementary Fig. 2 were provided as Supplementary Data 1 and Supplementary Data 2, respectively. The SARS-CoV-2 sequence data obtained for this study are openly available on European Nucleotide Archive (ENA) under the accession numbers ERR10442739, ERR10442746-47, ERR10442752-2801, ERR10442804, ERR10442809-10, ERR10442815-18, ERR10442825-28, and ERR10442833 (Project ERP142142). The data regarding demographic and clinical features related to each patient are available on request from the corresponding author. The data are not publicly available because they may contain information that could compromise privacy.

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

## Acknowledgements

The authors thank ANIA Foundation that financially supported this work. The authors also thank the whole staff of the Infectious Disease Unit of Azienda Ospedaliero-Universitaria Policlinico of Modena and of the Microbiology and Virology Laboratory of Bambino Gesù Children Hospital for outstanding technical support in samples collection, processing, and data management.

## Author contributions

C.A. conceived the study design and data analysis and wrote the manuscript; V.Fo., R.S. performed bioinformatic and biostatistical analyses, contributed to data interpretation and to manuscript writing; G.J.B., S.V., M.F. collected samples and data; V.Fi. and A.Gr. processed samples; V.C. contributed to bioinformatic analysis; G.J.B., M.F., S.V., S.E., A.Ga., B.F., E.F., M.M., A.C., S.B., A.V., P.B., C.R., G.G. recruited samples and enrolled patients; C.M. and C.F.P. conceived and directed the study, and critically revised the manuscript.

## Competing interests

We have no competing interest regarding the data reported in this paper. As general competing interests, C.F.P. acknowledges grants, boards, and sponsored lectures from Gilead, ViiV, Merck, Janssen, GSK, Astra Zeneca. C.M. has received research grants from Gilead and participated to advisory boards for Gilead, ViiV, Janssen, MSD, Angelini, Roche. C.A. acknowledges sponsored lectures from Pfizer. A.V. acknowledges research grants from MSD. S.B. has received research grants from Gilead. C.R. acknowledges research grants from DiaSorin. The remaining authors declare no competing interests.
