## [Peer Review File · Communications Biology]

Reviewers' comments:

Reviewer #1 (Remarks to the Author):

The authors tried to answer an intriguing question with this small but well-designed study. Given the premise that molnupiravir is mutagenic while Paxlovid is not, it is no surprise that molnupiravir is more mutagenic than Paxlovid over SARS-CoV-2 genome. Limitations related to small sample size are well-acknowledged. Maybe the authors should discuss the potential for synergy with other antivirals (either small molecule or mAbs) and the risks for immune escape or false negative N-based quick antigen assays. I also encourage adding to the discussion more recent literature about prevalence of Mpro mutations conferring resistance to Paxlovid.

Minor : "been authorised by EUA" should be change into "emergency use authorized by FDA, EMA and other regulatory authorities around the world" .

"small drugs therapies: should be changed into " small molecule antivirals"

Reviewer #2 (Remarks to the Author):

With great interest we studied the manuscript of Alteri et al., on the genomic evolution of SARS-CoV-2 in patients on antiviral therapy. We believe this is a very timely study as these antivirals are now widely used, and techniques to study their impact in the clinic are needed. These also include genomic techniques that can address resistance development. The authors collected a valuable cohort of patients and clinical samples and did a thorough analysis of the sequence information. The manuscript is well written and very informative and provides a sound platform for future studies.

Major comments:

1. At this moment the authors refer to ref 18 "Zhou et al" as the source of information regarding potential nirmatrelvir resistance associated mutations. The authors mention in line 300 that no selection of these "so far known" mutations occurred. We propose that the authors are more explicit as to which positions in Mpro they exactly investigated in their dataset and provide a list (L50, E166, A173, ...). In this way it becomes clear to the reader what has been exactly analyzed. A more complete source for substitutions potentially associated with nirmatrelvir resistance is Table 8 in the Paxlovid label (<https://www.fda.gov/media/155050/download>).

2. In the discussion there is currently a hypothesis from line 333-347. We feel this is difficult to understand. Perhaps the authors should instead consider something like "More experiments are needed to firmly establish if these substitutions really evolved under selection pressure and what the mechanisms are." Also for the hypothesis in line 348-356 the authors refer to a preprint from 2020 which is therefore perhaps not a strong source to build a hypothesis on.

Minor comments:

1. not all readers are familiar with the concept of genetic distance/diversity/variation. Perhaps keep one term throughout the paper (examples distance) and in the methods mention that these numbers are always expressed as 'nucleotide substitutions per site'. Also in each figure or table where these numbers are shown indicate that they have units 'nucleotide substitutions per site'. It would also be best to mention this unit of genetic distance in the abstract.

line 81: effects->effect

line 84: remove 'other'

line 96: include reference to paper Wong et al 2022 Lancet Infect Dis. 2022 Aug 24:S1473-3099(22)00507-2. doi: 10.1016/S1473-3099(22)00507-2.

line 106: include ref:

- bioRxiv 2022.08.07.499047; doi: <https://doi.org/10.1101/2022.08.07.499047>

- bioRxiv 2022.06.07.495116; doi: <https://doi.org/10.1101/2022.06.07.495116>

line 119: include city and country of hospital

line 141: was the swab first incubated in a buffer? what volume?

line 146: "a 50" -> "50"

line 177: remove "each"

line 213: "Among differences among" ...?

line 214 and other places: replace "respect to" with "compared to"

supplement fig 2: what is the unit of prevalence on the axis? is this nucleotide substitution per site divided by the protein length? please make clear in legend, why protein length and not nucleotide sequence length?

Ref. No.: COMMSBIO-22-3154A

Title: GENOMIC EVOLUTION OF SARS-COV-2 IN MOLNUPIRAVIR-TREATED PATIENTS COMPARED TO PAXLOVID-TREATED AND DRUG-NAÏVE PATIENTS: A PROOF-OF-CONCEPT STUDY

Journal: Communications Biology

We would like to thank the reviewers for their time and for appreciating our work.

The revisions of the manuscript in accordance with their comments are summarized in our responses below.

Response to Reviewers' Comments

Reviewer #1

Remarks to the Author:

The authors tried to answer an intriguing question with this small but well-designed study. Given the premise that molnupiravir is mutagenic while Paxlovid is not, it is no surprise that molnupiravir is more mutagenic than Paxlovid over SARS-CoV-2 genome. Limitations related to small sample size are well-acknowledged.

Answer: We thank the reviewer for his/her positive and thoughtful comments.

Major comments:

1) Maybe the authors should discuss the potential for synergy with other antivirals (either small molecule or mAbs) and the risks for immune escape or false negative N-based quick antigen assays.

Answer: Following reviewer suggestion, we have added a paragraph in the discussion about the potential synergistic effect of the combination therapy, lines 383-389, as follows: "Another aspect that should be taken into consideration is the use of Molnupiravir in combination with Paxlovid- or monoclonal antibodies- based therapy. Since the mode of action of these molecules is different, combined therapy might represent an advantage compared to monotherapy, allowing to boost the efficacy against SARS-CoV-2 infection and to reduce the risk of drug resistance or immune escape mutations. Studies reporting a synergistic effect of Molnupiravir and Paxlovid combination strategy both in vitro (52, 53) and in animal models (54) are already published, but ad hoc designed clinical studies will be needed to confirm these findings in humans." Three references (52-54) were also added. We also rephrased the sentence in lines 376-379 on antigenic test, as follows: "Regarding the antigenic tests, false negative results might be considered unlikely events under Molnupiravir

treatment, because of the rare persistence of non-synonymous mutations and the absence of positive selection in the target genes of antigen tests.”

2) I also encourage adding to the discussion more recent literature about prevalence of Mpro mutations conferring resistance to Paxlovid.

Answer: Following the Reviewer 2 suggestion, we have added more recent references about Mpro mutations in the introduction, results and discussion sections (ref 20,21,43).

Minor comments:

1) Minor: "been authorised by EUA" should be change into "emergency use authorized by FDA, EMA and other regulatory authorities around the world"
"small drugs therapies: should be changed into " small molecule antivirals"

Answer: Correction in the text have been made accordingly, in lines 64-65, 316 and 412.

Reviewer #2

Remarks to the Author:

With great interest we studied the manuscript of Alteri et al., on the genomic evolution of SARS-CoV-2 in patients on antiviral therapy. We believe this is a very timely study as these antivirals are now widely used, and techniques to study their impact in the clinic are needed. These also include genomic techniques that can address resistance development. The authors collected a valuable cohort of patients and clinical samples and did a thorough analysis of the sequence information. The manuscript is well written and very informative and provides a sound platform for future studies.

Answer: We thank the reviewer for her/his positive and thoughtful comments.

Major comments:

1) At this moment the authors refer to ref 18 "Zhou et al" as the source of information regarding potential nirmatrelvir resistance associated mutations. The authors mention in line 300 that no selection of these "so far known" mutations occurred. We propose that the authors are more explicit as to which positions in Mpro they exactly investigated in their dataset and provide a list (L50, E166, A173, ...). In this way it becomes clear to the reader what has been exactly analyzed. A more complete source for substitutions potentially associated with nirmarelvir resistance is Table 8 in the Paxlovid label (<https://www.fda.gov/media/155050/download>).

Answer: The entire Mpro protein was investigated and the mutated positions have been added as requested (302-305), as follows: "No evidence of so far known (19,43) Mpro mutations conferring resistance to Paxlovid was found. Only the following Mpro mutations were found: C10301T-Q83*

found at Day 7 of patient 4 with a relative abundance of 44.6%, and T10504C-F150F found at Day 2 of patient 3 with a relative abundance of 15.1%.”

Two sentences were also added regarding a mutation observed in a Molnupiravir-treated patient (lines 305-308), as follows: “Interestingly, the mutation C10376T-P108S, whose emergence was observed during Nirmatrelvir treatment in SARS-CoV-2 cell culture, was observed at Day 5 of Molnupiravir treatment in the patient 11 (relative abundance of 14.5%). However, no data of reduction of Nirmatrelvir activity was reported for this mutation in biochemical assays (43).”

The reference provided by the reviewer has also been included in the manuscript (ref 43).

2) In the discussion there is currently a hypothesis from line 333-347. We feel this is difficult to understand. Perhaps the authors should instead consider something like "More experiments are needed to firmly establish if these substitutions really evolved under selection pressure and what the mechanisms are." Also for the hypothesis in line 348-356 the authors refer to a preprint from 2020 which is therefore perhaps not a strong source to build a hypothesis on.

Answer: the sentences have been changed to soften the message conveyed, according to reviewer’s suggestion (lines 331-342), as follows: Despite the intra-host SARS-CoV-2 variation, already observed among clinical samples throughout the pandemic (45,46), SARS-CoV-2 strains under the Molnupiravir and Paxlovid pressure, as well as in drug-naïve samples, did not seem to experience strong selective evolution. By calculating the probability to evolve under purifying selection of each gene, only orf8, known to be involved in immune response evasion (41), seemed to evolve under a positive selection during Molnupiravir treatment, as a consequence of the 7/7 non-synonymous mutations detected randomly along the protein. Among these, only the orf8 A27927G-T12A persisted between Day 5 and Day 7. No clear information is available about the role of this amino acid mutation in protein stability or functionality, but *in silico* prediction suggests that this mutation might induce a protein stability change (free energy change, $\Delta\Delta G$, of -1.49 Kcal/mol compared to wild type) (47). Further *in vitro* studies are needed to confirm these *in silico* findings as well as in general to firmly establish whether and how these substitutions evolve under selection pressure.

Minor comments:

1) Not all readers are familiar with the concept of genetic distance/diversity/variation. Perhaps keep one term throughout the paper (examples distance) and in the methods mention that these numbers are always expressed as 'nucleotide substitutions per site'. Also in each figure or table where these numbers are shown indicate that they have units 'nucleotide substitutions per site'. It would also be best to mention this unit of genetic distance in the abstract.

Answer: As suggested, “diversity” was substituted with “distance” in lines 37, 240, 247 and 320. Moreover, the ‘nucleotide substitutions per site’ units have been added in the abstract (line 38),

methods (line 163), and results section (line 248), as well as in Figure 2, Table 2 and Suppl. Table 1, as suggested.

2) line 81: effects→effect

Answer: Correction has been made accordingly (line 81).

3) line 84: remove 'other'

Answer: Correction has been made accordingly (line 84).

4) line 96: include reference to paper Wong et al 2022 Lancet Infect Dis. 2022 Aug 24:S1473-3099(22)00507-2. doi: 10.1016/S1473-3099(22)00507-2.

Answer: Reference has been added as reference n° 13.

5) line 106: include ref:

bioRxiv 2022.08.07.499047; doi: <https://doi.org/10.1101/2022.08.07.499047>

bioRxiv 2022.06.07.495116; doi: <https://doi.org/10.1101/2022.06.07.495116>

Answer: References have been added as references n° 20 and 21.

6) line 119: include city and country of hospital

Answer: City and Country of Hospital have been added in lines 118-119.

7) line 141: was the swab first incubated in a buffer? what volume?

Answer: The nasopharyngeal swab, its buffer and volume have been added to lines 141-142, as follows: "(COPAN FLOQSwab with 3mL of Universal Transport Medium)"

8) line 146: "a 50" -> "50"

Answer: Correction has been made accordingly in line 147.

9) line 177: remove "each"

Answer: Correction has been made accordingly in line 177.

10) line 213: "Among differences among" ...?

Answer: It was changed to 'Among' in line 214.

11) line 214 and other places: replace "respect to" with "compared to"

Answer: Corrections have been made accordingly in lines 45, 215, and 216.

12) Supplement fig 2: what is the unit of prevalence on the axis? is this nucleotide substitution per site divided by the protein length? please make clear in legend, why protein length and not nucleotide sequence length?

Answer: We thank the reviewer for noticing our mistake on constructing the graph on protein length instead of nucleotide length. We have re-constructed the graph based on nucleotide substitutions per site divided by nucleotide length and reported the units in the legend as suggested. The figure legend is now the following: Supplementary Figure 2. Prevalence of C to T and G to A transitions (A) and other nucleotide substitutions (B) per site divided by nucleotide sequence length. Graph bars are colored based on study group.

REVIEWERS' COMMENTS:

Reviewer #1 (Remarks to the Author):

I have no further criticism and recommend prompt acceptance to inform public health choices.

Reviewer #2 (Remarks to the Author):

We feel the authors have replied to all comments proposed after the first review. We do not have additional comments on this manuscript and believe that the content is ready for publishing.

One small thing that could be improved is perhaps using always the same numbers of significant digits. I believe the use of two significant digits would be appropriate. This is currently the case when reporting viral load decrease (example line 222: median [IQR] -4.3 [-2.4;-6.2] vs. -3.0 [-1.6;-3.7]).

Two significant digits could also be used for reporting genetic distance. Currently the authors use 3 or 4 digits but this does not improve reading (example line 251 "(18.66x10⁻⁴ ± 2.06x10⁻⁴)" would be better as (19x10⁻⁴ ± 2.1x10⁻⁴)).

Ref. No.: COMMSBIO-22-3154A

Title: A PROOF-OF-CONCEPT STUDY ON THE GENOMIC EVOLUTION OF SARS-COV-2 IN MOLNUPIRAVIR-TREATED, PAXLOVID-TREATED AND DRUG-NAÏVE PATIENTS

Response to Reviewer's comments:

Reviewer #1 (Remarks to the Author):

I have no further criticism and recommend prompt acceptance to inform public health choices.

Answer: We thank the Reviewer for the appreciation of our work

Reviewer #2 (Remarks to the Author):

We feel the authors have replied to all comments proposed after the first review. We do not have additional comments on this manuscript and believe that the content is ready for publishing.

One small thing that could be improved is perhaps using always the same numbers of significant digits. I believe the use of two significant digits would be appropriate. This is currently the case when reporting viral load decrease (example line 222: median [IQR] -4.3 [-2.4;-6.2] vs. -3.0 [-1.6;-3.7]).

Two significant digits could also be used for reporting genetic distance. Currently the authors use 3 or 4 digits but this does not improve reading (example line 251 "(18.66x10⁻⁴ ± 2.06x10⁻⁴)" would be better as (19x10⁻⁴ ± 2.1x10⁻⁴)).

Answer: We thank the Reviewer for the general comment and for noticing the difference in the significant digits reported throughout the text. To uniform the data, we have decided to round up all data to 1 decimal digit. We have thus modified the manuscript, as well as Table 2, Table 3, Supplementary Table 1, and Supplementary Figure 2.